# Falls in Persons with Cognitive Impairment—Incidence and Characteristics of the Fallers

**DOI:** 10.3390/geriatrics9060168

**Published:** 2024-12-22

**Authors:** Per G. Farup, Knut Hestad, Knut Engedal

**Affiliations:** 1Department of Research, Innlandet Hospital Trust, N-2381 Brumunddal, Norway; knut.hestad@inn.no; 2Department of Clinical and Molecular Medicine, Faculty of Medicine and Health Sciences, Norwegian University of Science and Technology, N-7491 Trondheim, Norway; 3The Norwegian National Center for Aging and Health, Vestfold Hospital Trust, N-3103 Tønsberg, Norway; knutengedal@outlook.com; 4Department of Geriatric Medicine, Oslo University Hospital, N-0424 Oslo, Norway

**Keywords:** cognitive impairment, dementia, falls

## Abstract

**Background/Objectives:** The annual incidence of falls is high in older adults with impaired cognitive function and dementia, and injuries have a detrimental effect on disability-adjusted life-years and public health spending. In this registry-based study, fall incidence and characteristics of the fallers were explored in a large population with cognitive impairment. **Methods**: NorCog, “The Norwegian Registry of Persons Assessed for Cognitive Symptoms”, is a national research and quality registry with a biomaterial collection. This study included 9525 persons from the registry who had answered the question about falls. Fall incidence was studied, and the characteristics of fallers and non-fallers were compared. **Results**: The annual fall incidence was 3774/9525 (39.6%). The incidence varied between types of dementia, from 22.4% in persons with the debut of Alzheimer’s disease before 65 years of age to 55.3% in persons with vascular dementia and with increasing degrees of cognitive impairment. A wide range of personal characteristics, symptoms, signs, laboratory tests, and physical, psychological, and cognitive tests differed between fallers and non-fallers, most in disfavour of the fallers. Age, reduced Personal Activities of Daily Living, reduced gait speed, delayed recall, use of a walking aid, and depression were independent predictors of falls. **Conclusions**: Among cognitively impaired persons with a history of falls, frailty was an independent predictor of falls. Neither the type of dementia nor the degree of cognitive impairment were independent predictors of falls. Prevention of frailty by physical training and social activity may be important in mitigating fall risk among older adults with impaired cognition.

## 1. Introduction

Falls and fall injuries are common, and the incidence increases with age [1]. An overview of disability-adjusted life-years (DALYs) by The Global Burden of Disease Study 2021 reports that falls were number 22 of 371 diseases and injuries in people independent of age. [2]. In Norway, fall injuries are number two on the list of disease-specific health spending, accounting for 4.6% of the total spending [3]. Most studies focus on fall-related injuries, but fortunately, approximately half of all falls are without significant injuries [4].

Also, the prevalence of cognitive impairment and dementia increases with age. In Norway, dementia had the highest health spending, with 10.2% of the total spending [3]. Dementia increases the risk of falls [1,5]. An annual incidence of 80% has been reported in older people with dementia, nearly double that of cognitively healthy [1]. The fall incidence varies between the types of cognitive impairment and dementia, from low in persons with mild cognitive impairment to high in Parkinson’s disease with dementia and dementia with Lewy bodies (DLB) [4].

The high incidence of falls and fall-related injuries in old adults with cognitive impairment deserves attention [6]. Prevention of falls and fall-related injuries could significantly improve the DALYs of this high-risk group and reduce the community’s health spending. Comprehensive world guidelines for fall prevention have been worked out [7]. Prevention is, however, demanding; even comprehensive multifactorial fall prevention programs fail [8]. The literature review by Minta et al. shows that fall risk factors vary across cognitive disorders and between studies [4]. The heterogeneous findings make the implementation of disease-specific prophylaxis difficult. Precise knowledge of disease-specific predictors of falls, such as person characteristics, cognitive function, actual and previous somatic and psychiatric comorbidity, physical fitness, and blood and cerebrospinal fluid analyses, is necessary for effective prophylaxis. This registry-based study aimed to explore the fall incidence and a wide range of characteristics of fallers in a large population with cognitive impairment.

## 2. Materials and Methods

### 2.1. Persons

The study used NorCog’s “The Norwegian Registry of Persons Assessed for Cognitive Symptoms” data, a national research and quality registry with a biomaterial collection [9]. From 2009 to 2021, the registry included 18,120 persons referred to outpatient clinics in Norwegian specialist healthcare units to assess cognitive symptoms and suspected dementia. Participation was voluntary and based on written informed consent. Only persons with the capacity to give informed consent were included.

### 2.2. Variables

An overview of all variables in the registry is given by Medbøen et al. [9]. This study used the following variables reported by the persons and their next of kin:

#### 2.2.1. Person Characteristics

Sex (male/female); age (years); height (cm); body weight (kg); body mass index (BMI: kg/m^2^).Education (years in formal education).Living alone (yes/no).Needing regular public health service (yes/no).Walking assistance: no; indoor or outdoor; indoor and outdoor (scores 0–2).Physical activity was calculated as the sum of two scores: Easy activity (not sweaty/breathless: none; <1 h/week, 1–2 h/week, and >3 h/week: scores 0–3) and strenuous activity (sweaty/breathless: none; <1 h/week, 1–2 h/week, and >3 h/week (scores 0, 3, 4, and 5). The sum score of easy and strenuous activity is 0–8.Gait speed: Walking 4 m. Short physical performance battery: (Impossible = 0; > 8.7 s = 1; 6.21–8.7 s = 2; 4.82–6.20 s = 3; < 4.82 s = 4) [10,11].Balance test. Short physical performance battery. Standing with the feet close together, side-by-side, parallel. The results are given in seconds. (normal > 10 s) [11,12].Personal Activities of Daily Living (PADL) were measured with the “Physical Self-Maintenance Scale” (Score 6–30) [13,14]. High scores mean less self-reliant.Instrumental Activities of Daily Living (IADL) was measured with the “Lawton Instrumental Activities of Daily Living Scale (Score 8–31) [13,14]. High scores mean less self-reliant.Systolic and diastolic blood pressure (mmHg) and heart rate (beats/min) sitting and standing for 1 and 3 min.

#### 2.2.2. Medical History, Cognitive Functions

Present or previous diseases were noted (yes/no): cerebrovascular disease; Parkinson’s disease; neurological disease; coronary heart disease; cardiac surgery; diabetes; cancer; chronic obstructive pulmonary disease (COPD); polyarthritis; depression (present).Cornell scale for depression in dementia (Score 0–38. No depression < 7; mild depression 7–11; moderate–severe depression ≥ 12) [15].MAYO Sleep questionnaire. Question 8 in the questionnaire rates the patient’s general level of alertness for the past 3 weeks from sleeping all day to normally awake (score 0–10) [16].MAYO composite fluctuation scale (Score 0–4. Dichotomized: normal 0–2 = 1; abnormal 3–4 = 1) [17].Duration of cognitive impairment (years).Mini-Mental State Examination (MMSE). Cognitive functions were evaluated with a validated Norwegian version of the commonly used MMSE screening test: score 0–30 [18,19]. Higher scores indicate better performance.∙ CERAD word list test immediate and delayed: 10 words. The immediate test was the sum of three recalls, and the delayed test was after 10 min. A demographically adjusted version of the word list test in the Consortium to Establish a Registry for Alzheimer’s Disease (CERAD) was used [20,21,22]. Higher scores indicate better performance.∙ Trail Making Tests A and B (TMT-A and TMT-B) measure mental flexibility. The results are given in seconds. Low scores are better performance.Drugs (number of drugs taken regularly).

#### 2.2.3. Laboratory Tests

Blood tests: Hb (g/100 mL); CRP (mg/L); SR (mm/1 h); Thrombocytes (10^9^/L); Creatinine (µmol/L); Albumin (g/L); Folic acid (nmol/L); Cholesterol (mmol/L); Homocystein (µmol/L); Na (mmol/L); K (mmol/L); Ca (mmol/L); Thyroxin (pmol/L); TSH (mU/L); Vitamin B12 (pmol/L); Vitamin D (nmol/L); HbA1c (nmol/mol); Methyl Malonic Acid (MMA: µmol/L); ALAT (U/L); Gamma-GT (U/L); ALP (U/L).Cerebro-spinal fluid: Phospho-tau (reference value < 80 pg/mL); Total-tau (reference values: age < 50: <300 pg/mL; age 50–70: <450 pg/mL; age > 70: <500 pg/mL); Betta-amyloid (reference value < 550 ng/L) [23].Genes: APOE alleles: E2E2; E2E3; E2E4; E3E3: E3E4; E4E4.

#### 2.2.4. Clinical Evaluation and Diagnoses

The final diagnoses were based on all available information, including biological biomarkers (blood and cerebrospinal fluid), MRI or CT of the brain, and FDG-PET, and discussed in interdisciplinary meetings before concluding.

Clinical evaluation: subjective cognitive impairment (SCI); mild cognitive impairment (MCI); dementia; other. The diagnoses of SCI and MCI were according to Jessen et al. [24] and Winblad et al. [25], respectively.Dementia diagnoses: Alzheimer’s Disease (AD) early debut (<65 years of age); AD late debut (≥65 years of age); mixed AD/Vascular dementia (VascD); VascD; frontotemporal dementia (FrTeD); DLB; unspecified dementia; unknown. The diagnoses of FrTeD and DLB were according to Neary et al. [26] and McKeith et al. [27], respectively. The other diagnoses were according to the ICD-10 criteria (the 10th revision of the International Statistical Classification of Diseases and Related Health Problems)

### 2.3. Statistics

Descriptive data are reported as number (proportion) and mean (SD). The chi-square test with the trend, if appropriate, and *t*-test were used for unadjusted comparisons between fallers and non-fallers. A multivariable logistic regression analysis with inclusion of age and sex, and a stepwise forward inclusion of other variables was used to adjust the comparisons between fallers and non-fallers. The normality of the residuals was assessed by inspection of the Q-Q-plot, and goodness of fit was assessed with the Hosmer–Lemeshow Goodness of Fit Test. Because of missing data, the number of persons in each analysis is given. *p*-values < 0.01 were judged as statistically significant because of numerous analyses. The analyses were performed with IBM SPSS Statistics for Windows, version 29.0 (IBM Corp., Armonk, NY, USA).

## 3. Results

### 3.1. Person Characteristics

Of the 18,120 persons in the registry, 9525 (53%) had answered the question about falls and were included in this study, of whom 3774 (39.6%) had a fall in the last 12 months. Table 1 gives the person’s characteristics.

### 3.2. Comorbidity

Somatic and psychiatric comorbidity were more prevalent in fallers than in non-fallers. Table 2 gives the details of previous and actual comorbidity, cognitive functions, and the actual use of regular drugs.

### 3.3. Analyses of Blood and Cerebrospinal Fluid and Genetic Tests

The analyses of the blood and cerebrospinal fluid specimens and the genetic tests varied significantly between the fallers and non-fallers. Table 3 gives the details.

### 3.4. Clinical Evaluation and Final Diagnosis

Table 4 gives the degree of cognitive impairment and the dementia diagnoses based on all available information with comparisons between fallers and non-fallers.

### 3.5. Independent Predictors of Falls

Age, physical fitness tests, activities of daily living, and cognitive functions were predictors of falls. The significant independent predictors of falls are given in Table 5.

## 4. Discussion

The annual incidence of one or more falls (39.6%) in this population of older adults with impaired cognitive function and dementia was high but not unexpectedly high. Similar findings have been reported in other studies, but the heterogenicity of the study populations makes comparisons between the studies unreliable. Hollinghurst et al. report a 30% annual incidence of falls in adults above 65 years old, with approximately twice the incidence in older people with dementia [1]. Several studies have reported an increased incidence of falls with increasing age and degree of cognitive impairment [1,5,6]. The increased incidence with age was seen in both the unadjusted and adjusted analyses. The association between falls and declined cognitive function was documented by a higher fall incidence in persons with dementia than in those with mild cognitive impairment, low scores for MMSE, CERAD 10-word immediate and delayed recall, and MAYO sleep scale, and high MAYO fluctuation score and TMT-A and -B scores. Four of eight controlled trials reported associations between falls and cognitive functions [28]. The review and meta-analysis by Sturnieks et al. recommended the TMT-B test, a measure of mental flexibility, as a simple fall risk screen [6]. In the present study, the CERAD delayed recall was the only cognitive test independently predicting falls. The TMT tests differed significantly between the fallers and non-fallers but were not independent predictors. The higher fall incidence in females, as seen in other reports, was not seen in this study [1].

In addition to cognitive functions, social situations and physical fitness were predictors of falls. Fallers often lived alone and needed public health services and walking assistance. Physical activity, walking speed, and balance were reduced in fallers. Balance and gait deficit were predictors of fall in nine and eight out of nine controlled trials, respectively [28]. Physical performance, such as gait speed and balance, are predictors of health and survival [11,29]. However, larger studies conclude that these tests have limited sensitivity (31–79%) and specificity (52–74%) in predicting falls [4,30]. The tests could be more reliable in cognitively healthy than cognitively impaired persons [31].

Frailty is associated with reduced Activity of Daily Living (ADL) and falls [32]. In this study, both personal and instrumental ADLs were associated with falls. PADL, such as getting dressed and maintaining personal hygiene, and IADL, such as housekeeping and travelling, require a certain level of physical fitness, explaining the increased risk of falls in persons with impaired ADL. Impaired ADL has been reported as a cause of falls in five out of nine controlled trials [28].

Cardiovascular autonomic dysfunction occurs in frail persons with cognitive impairment, in particular in persons with DLB [33]. The main symptom is orthostatic hypotension, often defined as a reduction in systolic and diastolic blood pressure of 20 mm and 10 mm Hg, respectively, within three minutes of standing. Other symptoms are feelings of faintness, dizziness, blurred vision, and REM sleep disorders. The symptoms increase the risk of falls. In this study, the statistically significant but numerically small drop in the blood pressure in fallers compared with non-fallers makes autonomic dysfunction a less likely cause of falls. Only two out of seven controlled trials have reported an association between orthostatic hypotension and falls [28]. The sleep disorder and high MAYO fluctuation score, typical for DLB, could never-the-less indicate some degree of autonomic dysfunction in the fallers [17].

This study confirms the high prevalence of comorbidity in persons with cognitive dysfunction. Reviews show that nearly 90% of fallers have at least one other health condition; the average is four comorbidities [34]. This study also shows a significantly higher incidence of falls in eight of ten registered comorbidities. The falls could be related to the disease with functional impairments or to the treatment. Polypharmacy occurs frequently and makes side effects challenging to foresee [35,36]. In this study, the number of regularly used drugs was high. Only the number of regularly used drugs was registered in this study because the specification of drugs and doses was inaccurate.

A wide range of blood and cerebrospinal fluid analyses and genetic tests showed significant differences between fallers and non-fallers. Most of the differences were in disfavour of the fallers, but the minor differences could not explain the higher rate of falling.

The fall incidence varied significantly among the dementia diagnoses, from 22.4% in persons with the debut of AD before the age of 65 years to 55.3% in persons with VascD. The different incidence rates between various dementia groups (Table 4) are probably not only due to the type of dementia but also to age and comorbidities. Young persons with AD have no or only a few comorbidities, whereas older adults with vascular dementia normally have several due to atherosclerosis. Other studies have also shown significant variations in the fall incidence between dementia groups. Compared with healthy, the incidence has varied from twice as often in persons with mild dementia to 20 times in persons with Parkinson’s disease dementia [4]. The differences might be due to disease-specific functional differences and not the degree of dementia.

The independent predictors of falls were age, reduced physical fitness (reduced gait speed and need for a walking aid), psychiatric/cognitive impairment (depression and cognitive reduction), and a combination of physical and cognitive deficits (reduced PADL). Neither the dementia diagnoses, the clinical evaluation of cognitive deficit, MMSE, TMT-B, nor somatic comorbidity were independent predictors of falls. These results showed that falls in persons with cognitive impairment/dementia depend entirely on the person’s physical and mental functions, seemingly unrelated to the diseases causing these symptoms. Reduced gait speed and the need for a walking aid (a measure of balance) are physical functions related to dementia and somatic disease. In addition, cognitive function was associated with falls. The discrepancy between the reduced CERAD delayed recall associated with falls in the bivariate analyses and the opposite in the multivariable analyses is due to the interaction between age and delayed recall. Age is associated with an increased risk of falls and reduced recall, which explains the reduced recall in the bivariate analyses. Adjusted for age and other predictors of falls, the recall was, for unexplained reasons, significantly better in fallers. MMSE and TMT-B were not associated with falls in the multivariable analyses. The findings could indicate that cognitive functions are less significant predictors of falls than physical fitness. PADL is probably a measure of both physical fitness and cognitive function. The Cornell scale, which measures depression last week, was associated with falls. An inattentive and unconcerned mood might lead to careless behaviour and falls in persons with depression.

The low number of persons in some of the analyses and the lack of information about missing data are weaknesses of the study. Avoidance of falls is crucial for the individual as well as the community. The high fall incidence in this and other studies clearly shows the importance of effective prophylaxis. Guidelines for prevention are available [7]. As also seen in this study, frailty is a known predictor of falls [5,37]. Prevention of frailty by physical and mental training is probably the best prophylaxis against both falls and cognitive impairment.

## 5. Conclusions

Falls were common in this population with cognitive impairment and dementia, and prevention is of major importance. A wide range of disorders were associated with falls: person characteristics, physical and cognitive performance, analyses of blood and spinal fluid, and diagnoses of cognitive, psychiatric, and somatic diseases. The main finding was that falls were associated with frailty and were unrelated to the diseases causing these functional disorders. The significant frailty predictors of falls were age, reduced gait speed, need for a walking aid, reduced PADL, and depression. Prevention of frailty may be an important factor in mitigating fall risk among older adults with impaired cognition.

## Figures and Tables

**Table 1 geriatrics-09-00168-t001:** Characteristics of persons with and without a fall in the last 12 months with comparisons between the groups. The results are given as a number (proportion) or mean (SD).

Variable	Number	Fall	No Fall	Statistics *p*-Value
Gender: male Female	4604 4921	1778 (38.6%) 1996 (40.6%)	2826 (61.4%) 2925 (59.4%)	0.054
Age (year)	9525	76.2 (8.5)	72.4 (9.8)	<0.001
BMI (kg/m^2^)	8980	25.7 (4.7)	25.7 (4.8)	0.737
Education (year)	8816	10.7 (3.5)	11.6 (4.0)	<0.001
Living alone: Yes No	3319 5979	1554 (46.8%) 2115 (34.4%)	1765 (53.2%) 3864 (64.6%)	<0.001
Need for public health service: Yes No	3041 6283	1750 (57.5%) 1930 (30.7%)	1291(42.5%) 4353 (69.3%)	<0.001
Walking assistance: No Indoor or outdoor Both in and outdoor	4870 695 834	1518 (31.2%) 393 (56.5%) 624 (74.8%)	3352 (68.8%) 302 (43.5%) 210 (25.2%)	<0.001
Physical activity (score 0–8)	8159	2.8 (2.4)	3.7 (2.5)	<0.001
SPPB Walking (order 0–4)	4063	3.1 (1.0)	3.5 (0.8)	<0.001
SPPB Balance (seconds)	2207	6.5 (6.0)	8.6 (7.2)	<0.001
PADL (score 6–30)	8929	8.9 (3.5)	7.1 (2.1)	<0.001
IADL (score 8–31)	9088	16.3 (6.2)	13.0 (5.3)	<0.001
Blood pressure (syst.) sitting (mmHg)	8079	144.2 (23.2)	145.5 (21.6)	0.010
Blood pressure (diastolic) sitting	8077	81.2 (12.6)	82.3 (11.9)	<0.001
Heart rate sitting (beats/min)	7998	73.6 (13.9)	72.4 (13.2)	<0.001
Blood pressure (systolic) standing 1 min	6309	138.7 (25.0)	140.5 (23.2)	0.004
Blood pressure (diastolic) standing 1 min	6309	80.6 (17.9)	82.2 (12.8)	<0.001
Heart rate standing 1 min	6176	79.0 (14.9)	78.3 (18.0)	0.137
Blood pressure (systolic) standing 3 min	6088	142.5 (24.8)	143.4 (22.6)	0.157
Blood pressure (diastolic) standing 3 min	6083	82.3 (13.6)	83.8 (12.4)	<0.001
Heart rate standing 3 min	4715	79.3 (24.1)	78.0 (14.8)	0.025

SPPB: Short Physical Performance Battery. PADL: Personal Activities of Daily Living. IADL: Instrumental Activities of Daily Living.

**Table 2 geriatrics-09-00168-t002:** Medical history, cognitive functions, and drug treatment in persons with and without a fall in the last 12 months, with comparisons between the groups. The results are given as a number (proportion) or mean (SD).

Variable	Number	Fall	No Fall	Statistics *p*-Value
Cerebrovascular disease: Yes No	2111 7414	1007 (47.7%) 2767 (37.3%)	1104 (52.3%) 4647 (62.7%)	<0.001
Parkinson’s disease: Yes No	330 9195	199 (60.3%) 3575 (38.9%)	131 (39.7%) 5620 (61.1%)	<0.001
Neurological disease: Yes No	1434 8091	711 (49.6%) 3063 (37.9%)	723 (50.4%) 5028 (62.1%)	<0.001
Coronary heart disease: Yes No	5164 4361	2264 (43.8%) 1510 (34.6%)	2900 (56.2%) 2851 (65.4%)	<0.001
Cardiac surgery: Yes No	439 9086	171 (39%) 3603 (39.7%)	268 (61%) 5483 (60.3%)	0.803
Diabetes: Yes No	1282 8273	606 (48.4%) 3168 (38.3%)	646 (51.6%) 5105 (61.7%)	<0.001
Cancer: Yes No	1058 8467	454 (42.9%) 3320 (39.2%)	604 (57.1%) 5147 (60.8%)	0.021
COPD: Yes No	621 8904	306 (49.3%) 3468 (38.9%)	315 (50.7%) 5436 (61.1%)	<0.001
Polyarthritis: Yes No	448 9077	210 (46.9%) 3564 (39.3%)	238 (53.1%) 5513 (60.7%)	0.002
Depression (actual) Yes No	900 8625	439 (48.8%) 3335 (38.7%)	461 (51.2%) 5290 (61.3%)	<0.001
Cornell depression (scale 0–38)	6444	7.6 (5.9)	5.7 (5.3)	<0.001
MAYO Sleep (scale 0–10)	1763	7.2 (2.4)	8.0 (2.2)	<0.001
MAYO Fluctuation (scale 0–4)	3379	1.6 (1.2)	1.2 (1.2)	<0.001
MAYO Fluctuation: Normal (score < 3) Abnormal (score ≥ 3)	2716 663	1083 (39.9%) 356 (53.7%)	1633 (60.1%) 307 (46.3%)	<0.001
Duration of cognitive impairment (years)	5665	3.3 (3.6)	3.0 (3.0)	<0.001
MMSE (score 0–30)	7366	22.9 (4.6)	23.6 (4.8)	<0.001
CERAD—immediate recall	7640	11.8 (4.9)	12.6 (5.4)	<0.001
CERAD—delayed recall	7568	2.3 (2.2)	2.5 (2.9)	<0.001
TMT-A (seconds)	8446	98 (65)	80 (60)	<0.001
TMT-B (seconds)	4793	205 (107)	168 (98)	<0.001
Drugs (number)	6453	5.2 (3.4)	3.7 (3.0)	<0.001

COPD: chronic obstructive pulmonary disease. MMSE: Mini-Mental State Examination. CERAD: Consortium to Establish a Registry for Alzheimer’s Disease. TMT: Trail Making Test.

**Table 3 geriatrics-09-00168-t003:** Blood and cerebrospinal fluid analyses and genetic tests in persons with and without falls in the last 12 months, with comparisons between the groups. The results are given as mean (SD) or number (proportion).

Analyses	Number	Fall	No Fall	Statistics *p*-Value
**Blood tests**				
Hb (g/100 mL)	6820	13.6 (1.7)	14.1 (2.1)	<0.001
CRP (mg/L)	6090	5.5 (10.1)	4.1 (7.3)	<0.001
SR (mm/1 h)	3883	14.8 (15.3)	11.4 (12.7)	<0.001
Thrombocytes (10^9^/L)	6355	249 (78)	243 (66)	0.003
Creatinine (µmol/L)	6907	86.6 (41.5)	82.4 (33.7)	<0.001
Albumin (g/L)	6313	41.6 (11.7)	42.5 (6.2)	<0.001
Folic acid (nmol/L)	5879	20.3 (14.6)	19.3 (13.4)	0.008
Cholesterol (mmol/L)	5506	5.0 (1.9)	5.1 (1.4)	<0.001
Homocysteine (µmol/L)	3973	16.0 (6.7)	14.6 (5.9)	<0.001
Na (mmol/L)	6795	140.4 (3.0)	140.9 (5.5)	<0.001
K (mmol/L)	6798	4.3 (0.4)	4.3 (0.4)	0.813
Ca (mmol/L)	5411	2.27 (0.36)	2.33 (0.33)	<0.001
Thyroxin (pmol/L)	6146	15.6 (4.0)	15.5 (3.0)	0.284
TSH (mU/L)	6572	1.87 (1.21)	1.84 (1.15)	0.376
Vitamin B12 (pmol/L)	6521	390 (273)	374 (218)	0.007
Vitamin D (nmol/L)	1635	72.5 (27.8)	72.4 (25.8)	0.967
HbA1c (mmol/mol)	1505	42 (12)	40 (10)	<0.001
Methyl Malonate (µmol/L)	2187	0.24 (0.24)	0.22 (0.13)	0.012
ALAT (U/L)	6725	21.9 (13.9)	24.2 (17.4)	<0.001
Gamma-GT (U/L)	4649	42.8 (56.8)	38.2 (56.2)	0.007
ALP (U/L)	5880	77.5 (35.3)	73.1 (26.5)	<0.001
**Spinal fluid**				
Phospho-tau (pg/mL)	1494	65.5 (42.0)	75.8 (60.9)	<0.001
Total-tau (pg/mL)	1493	472 (300)	529 (332)	0.002
Beta-amyloid (ng/L)	1495	748 (326)	757 (346)	0.657
**Genes**	1563			
APOE E2E2	2	1 (50%)	1 (50%)	Chi-Square
APOE E2E3	83	33 (40%)	50 (60%)	Pearson
APOE E2E4	43	18 (42%)	25 (58%)	*p* = 0.303
APOE E3E3	645	237 (37%)	408 (63%)	
APOE E3E4	613	204 (33%)	409 (67%)	Lin-by-lin *
APOE E4E4	177	52 (29%)	125 (71%)	*p* = 0.019

* Lin-by-Lin: Linear-by-linear is a chi-square test for trends.

**Table 4 geriatrics-09-00168-t004:** Clinical evaluation and diagnoses in persons with and without falls last year with comparisons between fallers and non-fallers.

Variable	Number	Fall	No Fall	Statistics *p*-Value
Clinical evaluation Subj cognitive impairment Mild cognitive impairment Dementia Other	8340 704 3020 3878 738	181 (26%) 1144 (38%) 1701 (44%) 308 (42%)	523 (74%) 1876 (62%) 2177 (56% 430 (58%)	Pearson <0.001 Lin-by-Lin * <0.001
Dementia diagnoses: AD early debut (<65 years) AD late debut (≥65 years) Mixed AD/VascD VascD FrTeD DLB Unspecified dementia Unknown	1690 85 657 249 208 42 80 247 122	19 (22.4%) 221 (33.6%) 126 (50.6% 115 (55.3%) 12 (28.6%) 44 (55.0%) 132 (53.4%) 71 (58.0%)	66 (77.6%) 436 (66.4%) 123 (49.4%) 93 (44.7%) 30 (71.4%) 36 (45.0%) 115 (46.6%) 51 (42%)	Pearson *p* < 0.001 Lin-by-Lin * *p* < 0.001

* Lin-by-Lin: Linear-by-linear is a chi-square test for trends.

**Table 5 geriatrics-09-00168-t005:** The most important (according to the Wald value) and statistically significant predictors of falls in persons with impaired cognitive functions. Logistic regression analysis with fall as dependent variable. The total number of persons was 1833; persons with a fall were 759 (41.4%).

Independent predictors	B	Wald	OR	95% CI	*p*-Value
Sex (male)	0.111	1.157	1.118	0.912; 1.369	0.282
Age (years) *	0.023	9.007	1.023	1.008; 1.039	0.003
PADL	0.121	23.753	1.129	1.075; 1.185	<0.001
Cornell depression	0.038	15.360	1.039	1.019; 1.059	<0.001
Gait speed	−0.289	18.893	0.749	0.658; 0.853	<0.001
Walking aid	0.460	25.605	1.585	1.326; 1.864	0.001
CERAD delayed recall *	0.070	8.589	1.073	1.023; 1.124	0.003

* Interaction Age x CERAD delayed recall: B = 0.007; 95%CI: 1.000; 1.013; *p* = 0.035. Hosmer–Lemeshow Goodness of Fit: Chi-square: 4.903; *p* = 0.768. PADL: Personal Activities of Daily Living. CERAD: Consortium to Establish a Registry for Alzheimer’s Disease.

## Data Availability

The national registry NorCog is responsible for the source data. The de-identified data files from persons with falls were transferred to Innlandet Hospital Trust Brumunddal, Norway, and stored on a server dedicated to research. The security follows the rules given by The Norwegian Data Protection Authority, P.O. Box 8177 Dep. NO-0034 Oslo, Norway. The data are available upon request to the author.

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
