# Peer review of "Falls in Persons with Cognitive Impairment—Incidence and Characteristics of the Fallers"

_geriatrics, 2024, doi:10.3390/geriatrics9060168_

Round 1

Reviewer 1 Report

Comments and Suggestions for Authors

Thank you for the opportunity to review your manuscript.

I have some concerns.

Please see below for details.

The 2024 edition of "The Global Burden of Disease Study" mentioned in the introduction has been published in The Lancet. We recommend that you cite the most recent edition.

Also, the earlier studies on falls in the introduction do not reflect the latest guidelines and meta-analyses. Please reconsider.

Please add basic information such as the Norwegian population, aging rate, and life expectancy to the methods section.

The description of the statistical analysis is inadequate. First, please indicate the statistical software and version used for the analysis. Second, please state the required sample size. Finally, please state whether or not a normality test was performed. Also, the treatment of missing data is unclear. Does this mean that a listwise method was used? If so, you should also explain why you did not impute missing values using multiple imputation. Finally, please indicate how you entered predictors in the logistic regression analysis. In the results section, please present the goodness of fit of the constructed logistic model.

Please state that the discussion and conclusions of this study are novel. The results obtained here have already been reported in many meta-analyses. Please clarify the differences with these findings and suggest future perspectives.

That's all.

Author Response

Response to Reviewer 1

Comment:

The 2024 edition of "The Global Burden of Disease Study" mentioned in the introduction has been published in The Lancet. We recommend that you cite the most recent edition.

Answer:

The reference has been changed to the most recent one. The one we referred to, published in 2020, had data separately for older people, which was more appropriate for this paper, but the reference has been changed as proposed by the reviewer, and the text has been changed accordingly (page 1, lines 35-36).

Comment:

Also, the earlier studies on falls in the introduction do not reflect the latest guidelines and meta-analyses. Please reconsider.

Answer:

Two references: “World guidelines for fall prevention…” published in 2022, and “Cognitive functioning and falls in older people: A systematic review and meta-analysis”, published in 2024, have been added to the introduction (references 6 and 7, page 2, lines 48 and 50). As far as we know, these papers represent the latest guidelines and meta-analyses.

Comment:

Please add basic information such as the Norwegian population, aging rate, and life expectancy to the methods section.

Answer:

The data the reviewer asks for are not easily available and are judged to be of limited relevance to the paper. The population we have studied is not representative of the Norwegian population, but representative of persons with cognitive impairment in Norway.

 Comment about the statistics.

The description of the statistical analysis is inadequate. First, please indicate the statistical software and version used for the analysis.

Answer:

The analyses were performed with IBM SPSS Statistics for Windows, version 29.0 (IBM Corp., Armonk, NY, USA). The information has been added (page 4, lines 151-152)

Second, please state the required sample size.

Answer:

No formal sample size calculation was performed. The study used all available persons in the registry. The number of subjects in each analysis varied from 9525 in the registry to 1833 with complete data in the multivariable analysis, and down to 42 persons with Frontotemporal dementia. The number of persons in each of the analyses was unknown when the study was planned, and to calculate sample size when the results are available gives no meaning.

Finally, please state whether or not a normality test was performed.

Answer:

The normality of the residuals was assessed by inspection of the Q-Q-plot. The plot showed a good approximation to normality. Added on page 4, lines 147-148.

Also, the treatment of missing data is unclear. Does this mean that a listwise method was used? If so, you should also explain why you did not impute missing values using multiple imputation.

Answer:

All available data for each analysis were used, and the number of persons in each analysis is reported in the paper. To impute the huge amount of missing data from a small sample (1833) to the total population (9525) gives little meaning. For a correct imputation, knowledge of the type of missing data (“Missing completely at random”, “Missing at random”, and “Missing not at random") is necessary (Sterne et al, BMJ doi: 10.1136/bmj.b2393). This information is not available. It has been added to the discussion that the low number of persons in some of the analyses and the lack of information about missing data are weaknesses of the study (page 10, lines 281-282).

Finally, please indicate how you entered predictors in the logistic regression analysis.

Answer:

The logistic regression analysis was performed with the inclusion of age and sex, and then a stepwise forward inclusion of other variables. This information has been added to the statistical method section (page 4, lines 145-146).

In the results section, please present the goodness of fit of the constructed logistic model.

Answer:

The goodness of fit was assessed with the Hosmer-Lemeshow Goodness of Fit Test, the most reliable goodness of fit test in SPSS. This information has been added to the paragraph “Statistics” (page 4, lines 148-149), and the result is given in the “Result” section, table 5 (page 8)

Reviewer 2 Report

Comments and Suggestions for Authors

This manuscript represents an observational analysis of self reported falls among 9525 subjects registered on the Norwegian registry of persons assessed for cognitive symptoms.

The data indicates an increased prevalence of falls with increased cognitive impairment and increased frailty with no significant differences between reported types of dementia.

The abstract suggests that the degree of cognitive impairment is not related to the falls prevalence. This does not appear to be accurate.

The abstract suggests that preventing frailty will help reduce falls risk. Please explain how to prevent frailty.

The abstract suggests that mental training may be a prophylaxis against cognitive impairment. Is there any evidence that this is true or is this an unrelated suggestion?

Methods:

The diagnosis of type of dementia may not be accurate. This is common in large data sets and very few of the subjects have any biomarkers available. I agree that the presence of comorbidities likely produces a confounding effect on the results. Recommend limiting the number of variables assessed. Consider MMSE as the Measure of cognitive decline, living alone as a large independent risk factor, and frailty markers: SBP, PADL, physical activity score, IADL, and need for walking assistance.

Suggestion for the conclusion: cognitively impaired subjects are at high risk for falls and may be appropriate candidates  for a falls risk assessment and safety evaluation for modifiable risk factors. There is literature evidence for addressing modifiable fall risk factors as well as enhancing fall efficacy which should be noted.

Montero-Odasso M, van der Velde N, Martin FC, Petrovic M, Tan MP, Ryg J, Aguilar-Navarro S, Alexander NB, Becker C, Blain H, Bourke R, Cameron ID, Camicioli R, Clemson L, Close J, Delbaere K, Duan L, Duque G, Dyer SM, Freiberger E, Ganz DA, Gómez F, Hausdorff JM, Hogan DB, Hunter SMW, Jauregui JR, Kamkar N, Kenny RA, Lamb SE, Latham NK, Lipsitz LA, Liu-Ambrose T, Logan P, Lord SR, Mallet L, Marsh D, Milisen K, Moctezuma-Gallegos R, Morris ME, Nieuwboer A, Perracini MR, Pieruccini-Faria F, Pighills A, Said C, Sejdic E, Sherrington C, Skelton DA, Dsouza S, Speechley M, Stark S, Todd C, Troen BR, van der Cammen T, Verghese J, Vlaeyen E, Watt JA, Masud T; Task Force on Global Guidelines for Falls in Older Adults. World guidelines for falls prevention and management for older adults: a global initiative. Age Ageing. 2022 Sep 2;51(9):afac205. doi: 10.1093/ageing/afac205. Erratum in: Age Ageing. 2023 Sep 1;52(9):afad188. doi: 10.1093/ageing/afad188. Erratum in: Age Ageing. 2023 Oct 2;52(10):afad199. doi: 10.1093/ageing/afad199. PMID: 36178003; PMCID: PMC9523684.

Soh SL, Tan CW, Thomas JI, Tan G, Xu T, Ng YL, Lane J. Falls efficacy: Extending the understanding of self-efficacy in older adults towards managing falls. J Frailty Sarcopenia Falls. 2021 Sep 1;6(3):131-138. doi: 10.22540/JFSF-06-131. PMID: 34557612; PMCID: PMC8419849.

Author Response

Response to Reviewer 2

This manuscript represents an observational analysis of self reported falls among 9525 subjects registered on the Norwegian registry of persons assessed for cognitive symptoms.

The data indicates an increased prevalence of falls with increased cognitive impairment and increased frailty with no significant differences between reported types of dementia.

The abstract suggests that the degree of cognitive impairment is not related to the falls prevalence. This does not appear to be accurate.

Answer:

Cognitive impairment/types of dementia are related to falls in the bivariate analyses, but not in the multivariate analyses. The limited space in the abstract renders nuances difficult. The abstract has been slightly rewritten. (page 1, lines 23-28) Important conclusions of the paper are that degrees of cognitive functions/types of dementia are not independent predictors of falls.

The abstract suggests that preventing frailty will help reduce falls risk. Please explain how to prevent frailty.

Answer:

Explaining how to prevent frailty is outside the scope of this paper, but the importance of effective prophylaxis has been added (page10, lines 283-285).

The abstract suggests that mental training may be a prophylaxis against cognitive impairment. Is there any evidence that this is true or is this an unrelated suggestion?

Answer:

It is correct, as stated by the reviewer, that the effect of mental training on cognitive impairment is uncertain. “Mental training” has been replaced by “social activity”, which has a documented effect (Page 1, line 28)

Methods:

The diagnosis of type of dementia may not be accurate. This is common in large data sets and very few of the subjects have any biomarkers available. I agree that the presence of comorbidities likely produces a confounding effect on the results. Recommend limiting the number of variables assessed. Consider MMSE as the Measure of cognitive decline, living alone as a large independent risk factor, and frailty markers: SBP, PADL, physical activity score, IADL, and need for walking assistance.

Answer:

The final diagnoses were based on all available information, including biological biomarkers (blood and cerebrospinal fluid), MRI or CT of the brain, and FDG-PET, and discussed in interdisciplinary meetings before concluding. The diagnoses of dementia may not be accurate, but the background for the diagnoses is well documented.

Limiting the number of variables based on prejudiced expectations could include bias in the results. We agree that including MMSE as the measure of cognitive decline is tempting and we were surprised to find that MMSE was not an independent predictor of falls. We prefer the inclusion of variables without expectations of the results.

Suggestion for the conclusion: cognitively impaired subjects are at high risk for falls and may be appropriate candidates  for a falls risk assessment and safety evaluation for modifiable risk factors. There is literature evidence for addressing modifiable fall risk factors as well as enhancing fall efficacy which should be noted.

Answer:

The reviewers' suggestions for the conclusion focus on fall risk assessment, safety evaluation, modifiable risk factors, and enhancing fall efficacy. These topics are essential but partly outside the scope of this paper. We prefer a conclusion about the incidence of falls and predictors of falls, which was the aim of the study. The conclusions of the abstract and the text have been somewhat changed to comply with the reviewer's comments.

Round 2

Reviewer 1 Report

Comments and Suggestions for Authors

I have read the authors' comments and the revised manuscript.

I believe that appropriate revisions have been made.

Author Response

Comment:

I have read the authors' comments and the revised manuscript. I believe that appropriate revisions have been made.

Answer:

Thanks!

Reviewer 2 Report

Comments and Suggestions for Authors

The authors have made incredible response to review her continence.  Some concern regarding the message to readers particularly clinicians persists.  With the authors be agreeable to these changes?

Insert into line 25: Among cognitively impaired individuals with history of falls, frailty was an independent predictor of falls.

Delete lines to 82-82

Substitute this text order something similar 4 lines 26-28 and 2 85-2 90:

Prevention of frailty may be an important factor to mitigate fall risk among older adults with impaired cognition.

Author Response

Comments and Suggestions for Authors

The authors have made incredible response to review her continence.  Some concern regarding the message to readers particularly clinicians persists.  With the authors be agreeable to these changes?

Insert into line 25: Among cognitively impaired individuals with history of falls, frailty was an independent predictor of falls.

Answer:

The sentence proposed by the reviewer has been added to the abstract (line 25)

Comment: 

Delete lines to 82-82

Answer:

There seems to be some confusion concerning the line numbers in different versions of the paper. Line 82 in the last version of the paper was information about the balance test. This line has not been changed. Does the reviewer want to delete lines 80-81 about the gait speed? We want to keep the lines because the information is necessary to understand the scoring of the gait speed from 0 to 4.

Comment:

Substitute this text order something similar 4 lines 26-28 and 2 85-2 90:

 Prevention of frailty may be an important factor to mitigate fall risk among older adults with impaired cognition.

Answer:

The sentence proposed by the reviewer has been added to the abstract (lines 27-28) and the conclusion (lines 290-291.)